# A Simple but Strong Baseline for Sounding Video Generation: Effective Adaptation of Audio and Video Diffusion Models for Joint Generation

## Abstract

In this work, we build a simple but strong baseline for sounding video generation. Given base diffusion models for audio and video, we integrate them with additional modules into a single model and train it to make the model jointly generate audio and video. To enhance alignment between audio-video pairs, we introduce two novel mechanisms in our model. The first one is timestep adjustment, which provides different timestep information to each base model. It is designed to align how samples are generated along with timesteps across modalities. The second one is a new design of the additional modules, termed Cross-Modal Conditioning as Positional Encoding (CMC-PE). In CMC-PE, cross-modal information is embedded as if it represents temporal position information, and the embeddings are fed into the model like positional encoding. Compared with the popular cross-attention mechanism, CMC-PE provides a better inductive bias for temporal alignment in the generated data. Experimental results validate the effectiveness of the two newly introduced mechanisms and also demonstrate that our method outperforms existing methods. The source code will be released upon acceptance.

## 1 Introduction

Diffusion models have made great strides in the last few years in various generation tasks across modalities including image, video, and audio (Yang et al., 2023). Although these models are often large-scale and require a huge amount of computational resources for training, several prior studies such as Stable Diffusion (Rombach et al., 2022) have made their trained models publicly available, which substantially accelerates the progress of research and development on generative models. However, these models have mainly focused on a single modality, and it is still challenging to construct a model that is capable of generating multi-modal data.

In this work, we focus on audio-video joint generation, which is also known as *sounding video generation* (Liu et al., 2023b). Although sounding videos are one of the most popular types of multi-modal data, their generation has been addressed by only a few recent studies (Liu et al., 2023b; Ruan et al., 2023; Tang et al., 2023) due to the extremely high difficulty of handling heterogeneous and high-dimensional data for generative modelling. This challenge makes the training of multi-modal generative models much more expensive than that of single-modal models, and it creates a barrier to the research and development of sounding-video generation technologies.

In this paper, we present a simple baseline method for sounding video generation. We utilize the latest generative models in both the audio and video domains, and our method effectively integrates these models for audio-video joint generation. Specifically, we basically train only additional modules introduced during model combination, which reduces the cost for training. To enhance alignment within a generated pair of audio and video, we introduce two novel mechanisms: timestep alignment and Cross-Modal Conditioning as Positional Encoding (CMC-PE). Experimental results with several datasets validate the effectiveness of these mechanisms and also demonstrate that the proposed method performs on par with or better than existing methods in sounding video generation in terms of video quality, audio quality, and cross-modal alignment.

## 2 BACKGROUND AND RELATED WORK

### 2.1 DIFFUSION MODELS

Diffusion models (Yang et al., 2023) are a family of generative models designed to generate data by reversing a diffusion process. Here, we briefly review one of the most popular types of diffusion models, called the denoising diffusion probabilistic model (Ho et al., 2020).

#### 2.1.1 BASICS

The forward diffusion process comprises $T$ timesteps, and any data is gradually corrupted into pure random noise as the timesteps progress. Specifically, data at timestep $t$, denoted as $\mathbf{x}_t$, is obtained from the following conditional distribution:

$$q(\mathbf{x}_t|\mathbf{x}_{t-1}) = \mathcal{N}(\sqrt{1-\beta_t}\mathbf{x}_{t-1}, \beta_t\mathbf{I}), \tag{1}$$

where $\{\beta_t\}_{t=1}^T$ is a set of hyperparameters for a noise schedule that determines the amount of noise to be added at each timestep. The diffusion process defined above allows directly sampling $\mathbf{x}_t$ given $\mathbf{x}_0$ as follows:

$$q(\mathbf{x}_t|\mathbf{x}_0) = \mathcal{N}(\sqrt{\bar{\alpha}_t}\mathbf{x}_0, (1-\bar{\alpha}_t)\mathbf{I}), \ i.e. \ \mathbf{x}_t = \sqrt{\bar{\alpha}_t}\mathbf{x}_0 + \sqrt{1-\bar{\alpha}_t}\epsilon, \tag{2}$$

where $\bar{\alpha}_t = \prod_{s=1}^t (1-\beta_s)$, and $\epsilon \sim \mathcal{N}(0, \mathbf{I})$.

A transition from $\mathbf{x}_t$ to $\mathbf{x}_{t-1}$ in the reverse process can be approximated to be Gaussian, when $\beta_t$ is sufficiently small. Diffusion models are trained to estimate its mean by predicting the noise contained in $\mathbf{x}_t$, as

$$q(\mathbf{x}_{t-1}|\mathbf{x}_t) = \mathcal{N}(\mu_\theta(\mathbf{x}_t, t), \sigma_t^2\mathbf{I}), \tag{3}$$

$$\mu_\theta(\mathbf{x}_t, t) = \frac{1}{\sqrt{1-\beta_t}}\left(\mathbf{x}_t - \frac{\beta_t}{\sqrt{1-\bar{\alpha}_t}}\epsilon_\theta(\mathbf{x}_t, t)\right), \ \sigma_t^2 = \frac{1-\bar{\alpha}_{t-1}}{1-\bar{\alpha}_t}\beta_t, \tag{4}$$

where $\epsilon_\theta$ represents the model with learnable parameters $\theta$ for the noise prediction. It is also well-known that we can sample $\mathbf{x}_{t-1}$ in a deterministic manner using DDIM (Song et al., 2020), as:

$$\mathbf{x}_{t-1} = \sqrt{\bar{\alpha}_{t-1}}\hat{\mathbf{x}}_{0|t} + \sqrt{1-\bar{\alpha}_{t-1}}\epsilon_\theta(\mathbf{x}_t, t), \tag{5}$$

$$\hat{\mathbf{x}}_{0|t} := \frac{\mathbf{x}_t - \sqrt{1-\bar{\alpha}_t}\epsilon_\theta(\mathbf{x}_t, t)}{\sqrt{\bar{\alpha}_t}}. \tag{6}$$

Equation (3) (or Eq. (5)) enables us to sample slightly restored data given noisy data at any timestep. Consequently, given a random Gaussian noise $\mathbf{x}_T$, we can generate data $\mathbf{x}_0$ by repeating this sampling procedure from $t = T$ to $t = 1$.

The model $\epsilon_\theta$ is trained by minimizing a mean squared error of the predicted noise defined by

$$\min_\theta \mathbb{E}_{\mathbf{x}, \epsilon, t} \left\| \epsilon_\theta(\sqrt{\bar{\alpha}_t}\mathbf{x} + \sqrt{1-\bar{\alpha}_t}\epsilon, t) - \epsilon \right\|^2, \tag{7}$$

where $t$ is sampled from a uniform distribution $\mathcal{U}(1, T)$.

#### 2.1.2 APPLICATION TO SINGLE-MODAL GENERATION

Diffusion models have demonstrated remarkable performance across various modalities, particularly in vision and audio domains. In the vision domain, the initial attempt was limited to generating low-resolution images (Ho et al., 2020), but it was soon extended to handle high-resolution images (Rombach et al., 2022; Saharia et al., 2022) and videos (Ho et al., 2022; Guo et al., 2023; Blattmann et al., 2023). To reduce the computational cost due to the high dimensionality of data, the latest diffusion models are often trained in the space of latent features (Rombach et al., 2022) obtained by an encoder such as VAE (Kingma & Welling, 2014) or VQGAN (Esser et al., 2021). A similar trend can be found in the audio domain: diffusion models were initially used to directly generate waveforms (Chen et al., 2020b; Kong et al., 2020) and were then extended to generate compressed representation or latent features of audio signals (Liu et al., 2023a; Huang et al., 2023). In this work, we utilize the latest publicly available models in both domains, specifically, Animate-Diff (Guo et al., 2023) and AudioLDM (Liu et al., 2023a), to efficiently construct an audio-visual generative model that is capable of jointly generating video and audio well aligned with each other.

## 2.2 AUDIO-VIDEO GENERATIVE MODELS

### 2.2.1 CROSS-MODAL CONDITIONAL GENERATION

Video-conditioned audio generation (V2A) has been extensively explored in the literature of audio-visual generative models. Pioneering works adopted regression models (Owens et al., 2016; Chen et al., 2020c; Ghose & Prevost, 2020) and GANs (Hao et al., 2018; Ghose & Prevost, 2022), but auto-regressive models (Iashin & Rahtu, 2021; Du et al., 2023) and diffusion models (Luo et al., 2023; Mo et al., 2023; Su et al., 2023; Comunità et al., 2024; Wang et al., 2024) have become popular choices recently due to their scalability and capability of generating diverse data. To apply these models for V2A, we additionally need a mechanism to feed video conditional information into audio generation models. Such cross-modal conditioning has typically been achieved by a cross-attention mechanism (Vaswani et al., 2017), where the conditional information is used to compute keys and values in the attention process. In this work, we propose a new module for the cross-modal conditioning that is simple but effective for obtaining higher alignment between modalities.

Compared with V2A, audio-conditioned video generation (A2V) has not been as intensely addressed in the literature, as high-quality video generation itself is already challenging. Given the success of large-scale autoregressive models (Weissenborn et al., 2020; Yan et al., 2021) and diffusion models (Ho et al., 2022; Guo et al., 2023; Blattmann et al., 2023) in video generation, audio-to-video generation has also been addressed by extending these models to accept audio conditions (Ge et al., 2022; Yariv et al., 2023; Zhang et al., 2024a). In this paper, we also extend existing diffusion models of video generation, but our goal is to enable joint generation of audio and video, which is substantially more challenging than audio-to-video. For this purpose, we propose a new mechanism to adjust timesteps across modalities during the generation process. It is particularly required for joint generation to effectively handle noisy multi-modal data at each timestep for higher alignment, because any clean data is not accessible during the generation process differently from the situation of V2A or A2V.

### 2.2.2 AUDIO-VIDEO JOINT GENERATION

As mentioned above, audio-video joint generation, namely, sounding video generation, is challenging compared to single-modal generation, and few studies have tackled it (Liu et al., 2023b; Ruan et al., 2023; Tang et al., 2023). SVG-VQGAN (Liu et al., 2023b) adopts a novel tokenizer for audio and video to obtain suitable representation for multi-modal generation with auto-regressive models. MM-Diffusion (Ruan et al., 2023) and TAVDiffusion (Mao et al., 2024) are multi-modal diffusion models specifically designed for audio-video paired data. CoDi (Tang et al., 2023) integrates several single-modal dedicated diffusion models and additionally adopts environment encoders to extract modality-specific features to condition the generation process in the other modalities. All these models incur a large computational cost for training due to their new model architectures specialized for the joint generation. In this work, we aim to construct sounding video generation models with minimal effort by effectively transferring state-of-the-art models in both the audio and video domains. A very recent work by Xing et al. (2024) shares a similar motivation to ours, but it adopts guidance based approach, which heavily limits the capability of the model to generate temporally aligned samples. In contrast, we introduce an efficient adaptation method that significantly enhances temporal alignment across modalities.

## 3 PROPOSED METHOD

In this section, we first show an overview of our method and briefly explain how it works. Then, we describe the details of the two newly introduced mechanisms designed for boosting alignment between generated video and audio.

### 3.1 OVERVIEW

Our goal is to build a single model capable of generating video and audio jointly, utilizing two pre-trained diffusion models, one for video and the other for audio. These models, referred to as *base models*, are each represented by a U-Net structured neural network with pre-trained parameters. Our model, as shown in Figure 1, includes two U-Nets as base models with pre-trained modules depicted

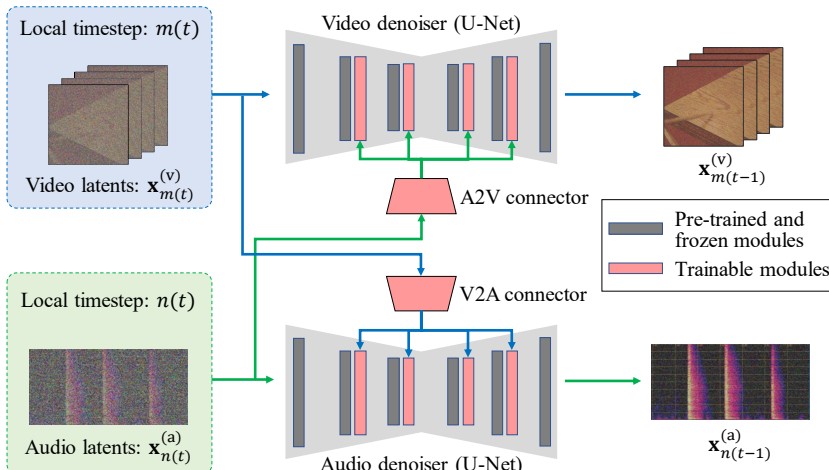

Figure 1: Overview of proposed model. For brevity of the diagram, we omit encoders to obtain latent features and paths for textual conditioning from both base models.

by gray rectangles. To enable joint generation of aligned video and audio, self-attention blocks are inserted into each U-Net, and *connectors* are introduced to extract features at each modality. These features are then fed into the U-Net of the other modality, allowing the model to utilize all modal information for noise prediction, resulting in better alignment across modalities, which will be described in Section 3.3. Note that, in the experiments, the noise prediction is conditioned by a given text prompt, as we used text-conditional generative models as the base ones. The text condition is fed into each U-Net in the same way as the original base models, and the same text prompt is used for both modalities.

During training, only the newly introduced modules are updated, while the pre-trained modules of each U-Net remain fixed. Like standard latent diffusion models, our model predicts noise from a pair of noisy latents and outputs slightly denoised latents. A key difference lies in the timestep setting, where different timesteps are set for each modality. This is due to the original design of the timestep in the U-Net at each modality, which may not be suitable for multi-modal joint generation. This issue and our solution are discussed in more detail in Section 3.2.

## 3.2 TIMESTEP ADJUSTMENT

### 3.2.1 WHY DO WE NEED TO ADJUST TIMESTEPS ACROSS MODALITIES?

The necessity of the timestep adjustment stems from a discrepancy in the noise schedules between modalities. As described in Eq. (1), the timestep information is closely related to the noise schedule $\{\beta_t\}$, and this schedule is pre-determined at each modality in our setting. Therefore, how samples are collapsed as the timestep progresses (or equivalently, how samples are generated as the timestep reverts) is not necessarily aligned between modalities.

To visualize this discrepancy, we plot the loss distribution over the timestep in Fig. 2a. The loss values are measured in the experiment (described in Section 4.1) and are normalized by their value at $t = 0$ for each modality. Here, we choose the loss instead of signal-to-noise ratio (SNR) as a proxy to observe how samples are generated, as SNR is not suitable for comparisons between data with a different number of dimensions (Hoogeboom et al., 2023). Obviously, the loss on video data is heavily skewed towards $t = 0$, which implies that the noise schedule for videos is set to more rapidly collapse data into noise as the timestep progresses. Such a noise schedule is often adopted to address the high dimensionality of video data (Hoogeboom et al., 2023). However, if we directly re-use this schedule in the joint generation setting, video information will not be very informative for audio generation at the intermediate timesteps, which makes the generation process more like audio-to-video than joint generation. To solve this problem, we need to adjust the timesteps across modalities.

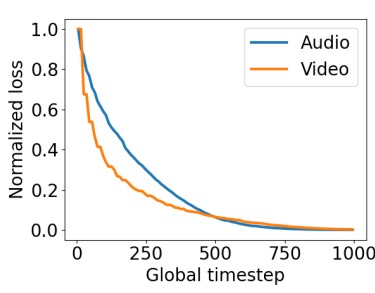

(a) Without timestep adjustment     (b) With timestep adjustment

Figure 2: Loss distribution over timesteps. The timestep adjustment makes the distributions closer to each other, which indicates that how samples are generated along with timesteps becomes more aligned across modalities after the adjustment.

### 3.2.2 A SIMPLE SOLUTION FOR TIMESTEP ADJUSTMENT

We adopt a global timestep $t$ and local timesteps, denoted by $m(t)$ and $n(t)$, for video and audio modality, respectively. The global timestep is set to control the noise level of all modalities and is evenly sampled in the generation process as usual timesteps. On the other hand, the local timesteps are set to adjust the noise level of each modality for higher alignment. We introduce a simple strategy to set the local timesteps, as follows:

$$m(t) = \text{round}\left(T_{\text{v}}\left(\frac{t}{T}\right)^{\sqrt{\gamma}}\right), \qquad n(t) = \text{round}\left(T_{\text{a}}\left(\frac{t}{T}\right)^{\frac{1}{\sqrt{\gamma}}}\right), \qquad (8)$$

where $\gamma$ is a hyperparameter for the timestep adjustment, and $T_{\text{v}}$ and $T_{\text{a}}$ are the maximum timestep in the base video and audio models, respectively. This definition is designed to make $m(t)/T_{\text{v}}$ proportional to $(n(t)/T_{\text{a}})^{\gamma}$. It means that, if we set larger $\gamma$, the local timestep in video generation is adjusted to be much smaller than that in audio generation. This leads to reducing the gap mentioned previously, while too large a $\gamma$ degrades the quality of the generated data due to the deviation from the original schedule (as we will show in the experiments). When $\gamma$ is set to one, both the local timesteps are set to be equal to the global timestep $t$, so nothing is adjusted in this setting.

Figure 2b shows the loss distributions after applying the adjustment with $\gamma = 1.5$. The horizontal axis represents the global timestep, and the vertical axis represents the normalized loss at each local timestep corresponding to the global one. Compared with Fig. 2a, the loss distributions become much more similar to each other. This indicates that how samples are generated along with the timestep becomes more aligned between video and audio. Consequently, through the joint generation, we can expect higher alignment between the generated pair of data. In this paper, we set $\gamma$ to 1.5, unless otherwise noted. How to automatically set this hyperparameter remains as future work.

### 3.3 HOW TO FEED CROSS-MODAL FEATURES INTO U-NET

#### 3.3.1 THE STANDARD CHOICE: CROSS-ATTENTION AND ITS LIMITATION

In the literature, the cross-attention mechanism has been extensively used for cross-modal conditioning in diffusion models. Figure 3a shows the simplest design in this approach adopted in our baseline method (Tang et al., 2023). For brevity, we discuss the case of audio-conditioned video generation, but it can also be applied to the case of video-conditioned audio generation. In this design, the conditional audio information is embedded into a single feature vector by an encoder, and it is used to compute keys and values in the cross-attention taken with the intermediate features in the video generation model. By training the encoder and the cross-attention block with audio-video paired data, we can make the model generate videos aligned with given audio information. Although this design is simple and widely applicable, it is quite challenging to achieve higher temporal consistency between the audio condition and generated video, since the single vector does not have sufficient capability to represent every piece of temporally local information in the audio condition.

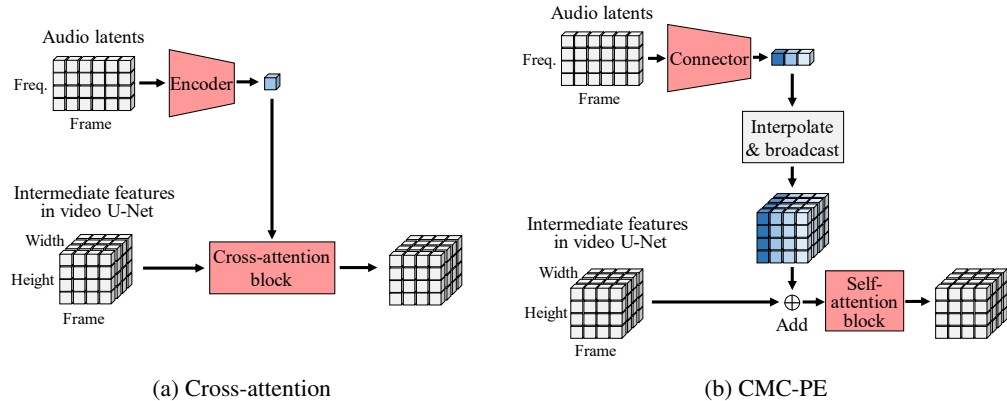

(a) Cross-attention         (b) CMC-PE

Figure 3: Mechanisms to feed conditional information into diffusion models. Each cube represents a single feature vector.

To boost the capability of the embedding features, we can extend the above-mentioned design by using multiple vectors each of which represents the temporally local information of conditional audio as done in Yariv et al. (2023). However, we still cannot strongly guarantee the temporal alignment, as it provides too much flexibility to connect the temporally-local audio information with the video to be generated, which may cause mis-alignment. It is also possible to adopt a more sophisticated attention mechanism (Ruan et al., 2023) or specifically dedicated encoder for embedding (Luo et al., 2023), but this substantially reduces applicability to the existing audio and video generation models, which does not fit our goal in this work.

### 3.3.2 CROSS-MODAL CONDITIONING AS POSITIONAL ENCODING (CMC-PE)

To achieve higher temporal alignment, we propose Cross-Modal Conditioning as Positional Encoding (CMC-PE), a simple but effective method of cross-modal conditioning. Figure 3b depicts how CMC-PE works. First, the conditional audio is encoded to a sequence of feature vectors along with time frames that work as if representing temporal position information. The extracted features are then added to the intermediate features in the video U-Net to function as positional embedding. To make this addition process valid, the features are interpolated and broadcast in advance to match their shape with that of the video features. Finally, the updated features are processed with a self-attention block. The features used for CMC-PE are extracted from current noisy latents $\mathbf{x}_t$ by the connector. We adopt the self-conditioning technique here (Chen et al., 2022), where the estimated data $\hat{\mathbf{x}}_{0|t}$ at each timestep shown in Eq. (6) is concatenated to the input.

CMC-PE has several advantages as follows. First, as the audio information is embedded into a sequence of vectors arranged in the time-frame direction, CMC-PE can utilize temporally local information that is suitable to temporally align the generated video with the conditional audio. Second, it has a strong inductive bias for higher temporal alignment, as the extracted temporally-local audio information is explicitly tied to the corresponding temporally local video information. Lastly, it is widely applicable to existing model architectures and conditional generation tasks. Once a target axis or axes of the intermediate features for desired alignment across modalities are given, CMC-PE can be extended in a straightforward manner.

### 3.4 TRAINING AND INFERENCE

Our model predicts the noise contained in the input pair of noisy latents $(\mathbf{x}_{m(t)}^{(v)}, \mathbf{x}_{n(t)}^{(a)})$, where $\mathbf{x}_{m(t)}^{(v)}$ and $\mathbf{x}_{n(t)}^{(a)}$ represent noisy video and audio latents at the global timestep $t$, respectively. For the training of the model, we extend the usual objective shown in Eq. (7) to the multi-modal setting and slightly modify it to make the trained model work with the timestep adjustment. Specifically, we

define the loss function as

$$\min_{\theta} \mathbb{E}_{\mathbf{x}, t_{\mathrm{v}}, t_{\mathrm{a}}} \left[ \mathcal{L}_{\theta}^{(\mathrm{v})}(\mathbf{x}, t_{\mathrm{v}}, t_{\mathrm{a}}) + \mathcal{L}_{\theta}^{(\mathrm{a})}(\mathbf{x}, t_{\mathrm{v}}, t_{\mathrm{a}}) \right], \tag{9}$$

$$\mathcal{L}_{\theta}^{(s)}(\mathbf{x}, t_{\mathrm{v}}, t_{\mathrm{a}}) = \mathbb{E}_{\epsilon_s} \left\| \epsilon_{\theta}^{(s)}(\mathbf{x}_{t_{\mathrm{v}}}^{(\mathrm{v})}, \mathbf{x}_{t_{\mathrm{a}}}^{(\mathrm{a})}, t_{\mathrm{v}}, t_{\mathrm{a}}) - \epsilon_s \right\|^2, \tag{10}$$

where $s \in [\mathrm{v}, \mathrm{a}]$ indicates the modality where the loss is to be computed, and $t_{\mathrm{v}}$ and $t_{\mathrm{a}}$ represent the local timestep for video and audio, respectively. A key point here is that the local timesteps are independently sampled from a uniform distribution. Due to this, the loss is computed on all possible combinations of the local timesteps so that the trained model can handle the timestep adjustment with any value of $\gamma$ specified in the inference phase. During the training of the model, the connectors and the inserted self-attention blocks are optimized with audio-video paired data, while the pre-trained parameters of the other modules are fixed, with one small exception (discussed in the appendix A.2).

The generation process in our method is almost the same as that in usual diffusion models except for the setting of local timesteps. In each step of the generation process, the model predicts noise for video and audio latents following the local timestep setting. Once these noises are predicted, we can apply any inference technique, also called a "sampler" (Yang et al., 2023), to obtain $\mathbf{x}_{m(t-1)}^{(\mathrm{v})}$ and $\mathbf{x}_{n(t-1)}^{(\mathrm{a})}$ in the same manner as in the base models. In this paper, we use one of the most popular ones, DDIM (Song et al., 2020) shown in Eq. (5), for both modalities. The generation process in the proposed method is summarized in the appendix A.1.

## 4 EXPERIMENTS

We first show the experimental results with a dedicated dataset to confirm that the two newly introduced mechanisms, the timestep adjustment and CMC-PE, contribute to boosting the alignment between generated video and audio. After that, we present the results with two benchmark datasets to show the effectiveness of our method through a comparison with several existing methods.

### 4.1 EXPERIMENTS WITH A DEDICATED DATASET FOR EVALUATING TEMPORAL ALIGNMENT

#### 4.1.1 DATASET AND EVALUATION METRICS

We extended the GreatestHits dataset (Owens et al., 2016) for our experiments. It contains 977 videos of humans hitting various objects with a drumstick in the scene. As the hitting sound and motion are dominant in the video, this dataset is suitable for evaluating the temporal alignment between the generated video and audio. We created video captions using LLaVA-NeXT (Zhang et al., 2024b) and utilized them as text conditions in our method. The details of this process are described in the appendix A.3.

We evaluated the quality of the generated data from three perspectives: video quality, audio quality, and temporal alignment. For the former two, we used FVD (Unterthiner et al., 2018) and FAD (Kilgour et al., 2019), both of which are widely utilized in the literature. To measure how much the generated video and audio are aligned with each other in term of temporal dynamics, we used the AV-Align score proposed by Yariv et al. (2023). This score is defined as Intersection-over-Union between onsets detected from the audio and peaks obtained from the optical flow. It is especially useful to measure the temporal alignment in the GreatestHits dataset, as hitting sounds make clear onsets, and hitting motions have correlating and distinct peaks in the optical flow.

We slightly modified how to compute AV-Align score from the official implementation. Specifically, we tuned hyper-parameters of the optical flow estimation and those of the onset detection to accurately estimate hitting timing using annotated timestamps in the Greatest Hits dataset. In addition, we compute IoU after rewriting it with precision and recall to mitigate an issue caused by the difference of temporal resolution between video and audio. Details are provided in the appendix A.4.

#### 4.1.2 SETUP

We trained our model to generate four-second audio-video pairs. Each video comprises eight frames per second, and the size of each frame is $256 \times 256$. The sampling rate of the audio is 16 kHz. We followed the train/test split in the original GreatestHits dataset.

Table 1: Experimental results with the GreatestHits dataset.

| Method | FVD ($\downarrow$) | FAD ($\downarrow$) | AV-Align ($\uparrow$) |
|---|---|---|---|
| Cross-attention (same as CoDi (Tang et al., 2023)) | 379 | 2.35 | 0.250 |
| Our method without timestep adjustment ($\gamma = 1$) | 393 | 1.29 | 0.256 |
| Our method ($\gamma = 1.25$) | 387 | 1.32 | 0.257 |
| Our method ($\gamma = 1.50$) | 381 | **0.60** | **0.268** |
| Our method ($\gamma = 1.75$) | **374** | 0.61 | **0.268** |
| Our method ($\gamma = 2.00$) | 383 | 1.32 | 0.265 |

To investigate the advantage of CMC-PE and the timestep adjustment, we compared the following three methods:

1. One with the same setting as CoDi (Tang et al., 2023), in which cross-attention blocks are used for cross-modal conditioning.

2. One using CMC-PE instead of cross-attention blocks (our method with $\gamma = 1$).

3. One using both CMC-PE and the timestep adjustment (our method with $\gamma > 1$).

For the training, we used the Adam optimizer (Kingma & Ba, 2015) with a learning rate of 1e-5, and the batch size and the number of epochs were set to 16 and 1,000, respectively. For generation, we set the number of global timesteps $T$ to 25. We adopted classifier-free guidance (Ho & Salimans, 2021) at each modality and set the strength of the guidance to 7.5 and 2.5 for video and audio, respectively, which are the standard settings in the original base models.

### 4.1.3 RESULTS

Table 1 shows the evaluation results. Replacing cross-attention with CMC-PE improves the AV-Align score as well as FVD and FAD, which demonstrates that CMC-PE has a better inductive bias for temporal alignment than the cross-attention mechanism. The AV-Align score is further improved by conducting the timestep adjustment when generating data. This is achieved by making the generation process in both modalities mutually informative as discussed in Section 3.2. Meanwhile, using too large a $\gamma$ leads to degradation of the performance due to the deviation from the original noise schedule. Overall, the proposed method performs substantially better in terms of the cross-modal alignment than our baseline following the design in CoDi (Tang et al., 2023).

Figure 4 shows examples of the generated audio-video pairs. The top and middle rows show video frames and the magnitude of their optical flows, respectively, and the bottom rows depict the waveforms of the generated audios. We confirmed that the onsets in the generated audio align well with the motion of a drumstick in the generated video, which demonstrates the capability of our model to produce aligned audio-video pairs.

## 4.2 EXPERIMENTS WITH BENCHMARK DATASETS

### 4.2.1 DATASET AND EVALUATION METRICS

To compare the proposed method with existing methods, we conducted experiments with two popular benchmark datasets: Landscape (Lee et al., 2022) and VGGSound (Chen et al., 2020a). The Landscape dataset consists of 928 videos covering nine classes of natural scenes, while VGGSound is a substantially larger and more diverse dataset containing nearly 200K video clips covering about 300 sound classes. To enhance the data quality, we filtered 60K videos in which audio-video alignment is weak, as done in TempoToken (Yariv et al., 2023). In both datasets, we used the class names as the text conditions and trained our model to generate four-second audio-video pairs. The video comprises four frames per second, and the size of each frame is $256 \times 256$. The sampling rate of the audio is 16 kHz. Note that we changed fps from the previous experiments to follow the setting in the prior studies (Iashin & Rahtu, 2021; Luo et al., 2023; Yariv et al., 2023).

Differently from the previous experiments, we did not use AV-Align for the evaluation, as the videos in both Landscape and VGGSound often lack distinct motions highly correlating their audios. In-

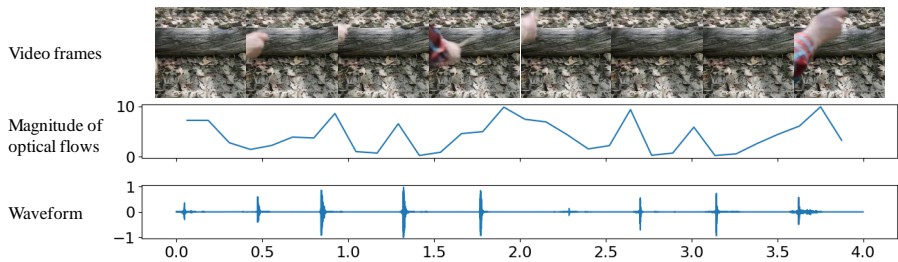

(a) "A person is hitting a drumstick on a log that is lying on the ground in a wooded area. The log is surrounded by fallen leaves and branches, and there are rocks and other debris in the background."

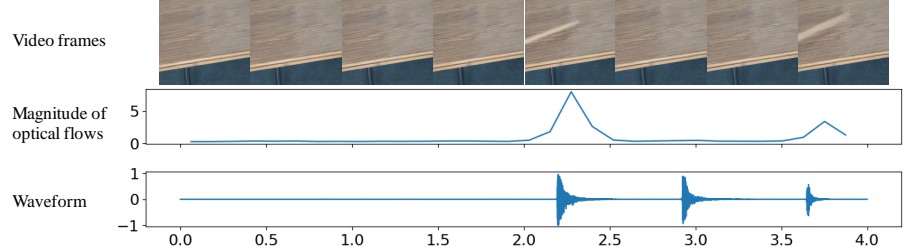

(b) "A person is hitting a table with a drumstick in the video. The table is a part of a piece of furniture with a flat surface and appears to be made of a material that can be struck with a drumstick."

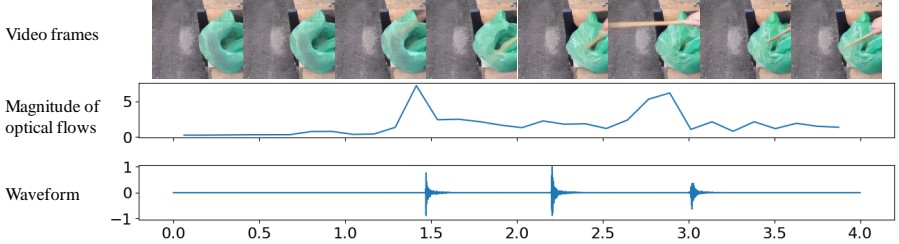

(c) "A person is hitting a drumstick against a blue plastic trash bag, which is placed on a wooden surface. The background is a wooden wall."

Figure 4: Examples of audio-video pairs generated by the proposed method.

stead, we used ImageBind score (Girdhar et al., 2023) between audio and video (IB-AV) to evaluate the cross-modal semantic alignment. Additionally, we computed the ImageBind score for text-audio and text-video pairs (IB-TA and IB-TV, respectively) to evaluate the audio and video quality in terms of fidelity to text condition.

### 4.2.2  SETUP

For comparison, we examined three approaches: text-to-audio + audio-to-video (T2A2V), text-to-video + video-to-audio (T2V2A), and audio-video joint generation. We chose several state-of-the-art generative models for each approach, as follows:

**T2A2V**  We used TempoToken (Yariv et al., 2023) to re-generate videos from the audios that are generated by the proposed method.

**T2V2A**  We used SpecVQGAN (Iashin & Rahtu, 2021) and DiffFoley (Luo et al., 2023) to re-generate audios from the videos that are generated by the proposed method.

**Joint generation**  We used MM-Diffusion (Ruan et al., 2023). For a fair comparison, the number of timesteps was set to be the same as that of the proposed method.

For all methods, we used the official implementation and pretrained models provided by the respective authors. Note that the pretrained models of SpecVQGAN and DiffFoley were available for

Table 2: Experimental results with Landscape dataset.

| Method | FVD ($\downarrow$) | IB-TV ($\uparrow$) | FAD ($\downarrow$) | IB-TA ($\uparrow$) | IB-AV ($\uparrow$) |
|---|---|---|---|---|---|
| TempoToken (Yariv et al., 2023) | > 3000 | 0.220 | – | – | 0.146 |
| MM-Diffusion (Ruan et al., 2023) | 1689 | – | 16.4 | – | 0.191 |
| Proposed method | **1122** | **0.238** | **6.63** | 0.146 | **0.192** |

Table 3: Experimental results with VGGSound dataset. († DiffFoley uses a larger dataset for learning cross-modal alignment)

| Method | FVD ($\downarrow$) | IB-TV ($\uparrow$) | FAD ($\downarrow$) | IB-TA ($\uparrow$) | IB-AV ($\uparrow$) |
|---|---|---|---|---|---|
| TempoToken (Yariv et al., 2023) | 2473 | 0.155 | – | – | **0.168** |
| SpecVQGAN (Iashin & Rahtu, 2021) | – | – | 5.08 | 0.059 | 0.100 |
| DiffFoley (Luo et al., 2023) | – | – | 5.72 | 0.074 | 0.159$^{\dagger}$ |
| Proposed method | **333** | **0.277** | **1.46** | **0.129** | 0.155 |

VGGSound, and that of MM-Diffusion was available for Landscape. When we cannot specify frame rate or resolution of the generated data, we resized the generated data to make it match our setting before the evaluation. For the training of our model, the batch size and the number of epochs were set to 16 and 100 for the Landscape dataset, and to 128 and 30 for VGGSound, respectively. The other settings are the same as those in the previous experiments.

### 4.2.3 RESULTS

Tables 2 and 3 show the results with Landscape and VGGSound, respectively. Firstly, TempoToken failed to generate high-quality videos, which indicates that it is not robust against even small artifacts in the conditional audio caused by preceding text-to-audio generation. This is one of the most critical issues of sequential approaches like T2A2V and T2V2A, and SpecVQGAN and DiffFoley also suffered from it, resulting in relatively low audio quality. In contrast, the proposed method achieved the best quality in both video and audio except for FVD in Landscape and IB-AV in VGGSound while attaining high cross-modal alignment. This indicates the importance of the training dedicated to joint generation and the effectiveness of our method.

### 4.3 LIMITATION

In the experiments, we observed that our model occasionally ignores some parts of the textual condition. For example, in Fig. 4c, while the conditional text contains "*a blue plastic trash bag*," the word *blue* is somewhat ignored in the generated video. We conjecture that this is caused by focusing too much on aligning with the audio to be jointly generated that do not contain visual information (e.g. color). How to simultaneously achieve fine-grained cross-modal alignment and semantic alignment with the textual condition would be an interesting avenue for future work. Additionally, as our method leverages pre-trained models, its performance heavily depends on that of these base models.

## 5 CONCLUSION

In this paper, we have built a simple but strong baseline method for sounding video generation. For efficient training, we only add small modules to a pair of existing audio and video diffusion models and train them with audio-video paired data for joint generation. In our method, we introduced two novel mechanisms, timestep adjustment and CMC-PE, to boost cross-modal alignment of the generated data. The timestep adjustment provides a modality-wise timestep schedule to align the speed at which samples are generated along with the timesteps at each modality. CMC-PE provides a better way to feed each modal feature into another-modal diffusion model in terms of inductive bias for higher temporal alignment compared with a popular cross-attention mechanism. The experimental results demonstrated that our method achieves high cross-modal alignment as well as high quality of the generated video and audio.

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

---

**Algorithm 1** Generation process in proposed method.

---

**Require:** $\epsilon_\theta$
  Initialize $\mathbf{x}_{m(T)}^{(v)}$ and $\mathbf{x}_{n(T)}^{(a)}$ with Gaussian noise.
  **for** $t$ in $[T,...,1]$ **do**
      Set local timesteps $m(t)$ and $n(t)$ using Eq. (8).
      $(\epsilon^{(v)}, \epsilon^{(a)}) \leftarrow \epsilon_\theta(\mathbf{x}_{m(t)}^{(v)}, \mathbf{x}_{n(t)}^{(a)}, m(t), n(t))$.
      Sample $\mathbf{x}_{m(t-1)}^{(v)}$ based on $\epsilon^{(v)}$ using Eq. (5).
      Sample $\mathbf{x}_{n(t-1)}^{(a)}$ based on $\epsilon^{(a)}$ using Eq. (5).
  **end for**
  **Return** $\mathbf{x}_0^{(v)}$ and $\mathbf{x}_0^{(a)}$.

---

# A APPENDIX

## A.1 GENERATION PROCESS IN THE PROPOSEED METHOD

The generation process in the proposed method is summarized in Algorithm 1.

## A.2 IMPLEMENTATION DETAILS

We used AnimateDiff (Guo et al., 2023) and AudioLDM (Liu et al., 2023a) as base models for video and audio, respectively. In both models, we insert the additional module just after each of the last two down-sampling blocks and each of the first two up-sampling blocks. Consequently, our model contains four additional modules at each modality. Following CoDi (Tang et al., 2023), each module is implemented by a single Transformer decoder block, while a cross-attention layer is replaced with a self-attention one for CMC-PE. We also followed the architecture of the environment encoder in CoDi for our connector. The total number of parameters in the newly added modules is about 468M. This is nearly four times smaller than that of the U-Nets in AnimateDiff and AudioLDM, which exceeds 1.7B.

As mentioned previously, we basically optimize the newly added modules as well as the connectors while fixing the pre-trained parameters during training. However, we made one exception, namely, the motion layers in AnimateDiff, which are also fine-tuned during training. This is because these layers are dedicated to a certain frame rate and duration of videos (specifically, eight frames per second and two seconds), which do not necessarily match those of the training data (as shown in Section 4).

## A.3 CAPTIONING FOR GREATESTHITS DATASET

We created video captions of the GreatestHits dataset utilizing LLaVA-NeXT (specifically LLaVA-NeXT-Video-7B-DPO released by Zhang et al. (2024b)). The text prompt is set as shown below:

> *What is being hit with a drumstick in the video? Please answer concisely with a brief description of its background by beginning with "A person is hitting".*

## A.4 COMPUTATION OF AV-ALIGN SCORES

### A.4.1 DEFINITION

AV-Align score Yariv et al. (2023) is defined as Intersection-over-Union (IoU) between onsets detected from the audio and peaks obtained from the optical flow. Specifically, it is computed as

$$\text{AV-Align} = \frac{1}{2|\mathcal{A} \cup \mathcal{V}|} \left( \sum_{a \in \mathcal{A}} 1[a \in \mathcal{V}] + \sum_{v \in \mathcal{V}} 1[v \in \mathcal{A}] \right), \tag{11}$$

where $\mathcal{A}$ represents a set of the onsets detected from the audio signal, and $\mathcal{V}$ represents a set of the peaks in the optical flow. A peak is considered to be valid in the other modality, if any corresponding peak exists within a window of three frames.

### A.4.2 OFFICIAL IMPLEMENTATION

One issue in the computation of the AV-Align score is in evaluating $|\mathcal{A} \cup \mathcal{V}|$. As the temporal resolution of the audio is much higher than that of the video, a single peak in the video may have multiple corresponding peaks in the audio. In this case, there is no trivial way to count the number of elements in $\mathcal{A} \cup \mathcal{V}$ due to this one-to-many matching property. To avoid this problem, the official implementation adopts the following equation to compute the AV-Align score:

$$\text{AV-Align} \leftarrow \frac{c}{|\mathcal{A}| + |\mathcal{V}| - c}, \text{ where } c = \sum_{a \in \mathcal{A}} 1[a \in \mathcal{V}]. \tag{12}$$

However, when $|\mathcal{A}| > |\mathcal{V}|$, the computed score can exceed one, which is unreasonable considering the original definition of the AV-Align score.

### A.4.3 MODIFICATION

We modified the score computation so that it follows the original definition of the score. First, we rewrite IoU using precision and recall as

$$\text{IoU} = \frac{\text{Precision} \cdot \text{Recall}}{\text{Precision} + \text{Recall} - \text{Precision} \cdot \text{Recall}}. \tag{13}$$

Utilizing this rewritten equation, we can compute the AV-Align score as

$$\text{AV-Align} \leftarrow \frac{pr}{p + r - pr}, \tag{14}$$

$$\text{where } p = \frac{1}{|\mathcal{A}|} \sum_{a \in \mathcal{A}} 1[a \in \mathcal{V}], \ r = \frac{1}{|\mathcal{V}|} \sum_{v \in \mathcal{V}} 1[v \in \mathcal{A}]. \tag{15}$$

By computing the score in this way, we can obtain a normalized value that is reasonable as IoU while avoiding the previously mentioned issue.

