# OpenReview forum: "A Simple but Strong Baseline for Sounding Video Generation: Effective Adaptation of Audio and Video Diffusion Models for Joint Generation"
_ICLR.cc/2025/Conference — ICLR 2025 Conference Withdrawn Submission_

### Official Review · Reviewer_81aL · 2024-11-01

**Soundness:** 3
**Presentation:** 3
**Contribution:** 1
**Rating:** 5
**Confidence:** 3

**Summary:**

This paper presents a new baseline model for generating synchronized audio and video. The authors integrate base diffusion models for audio and video into a single model, enhancing the alignment between audio-video pairs with two novel mechanisms. The first mechanism, timestep adjustment, provides different timestep information to each base model to align sample generation across modalities. The second mechanism, Cross-Modal Conditioning as Positional Encoding (CMC-PE), embeds cross-modal information as temporal position information, similar to positional encoding.

**Strengths:**

1. Combining two pretrained single-modal diffusion models for multi-modal joint generation is an invaluable and interesting task, the author provides a potential effective solution to this task.

2. The author pointed out that the discrepancy in the noise schedules between modalities can damage the temporal alignment of both modals, which is insightful and proven effective.

**Weaknesses:**

1. Quantitative results can not support the effectiveness of CMC-PE. As demonstrated in Table.1, when using CMC-PE (the 2nd line) rather than the cross-attention (the 1st line), the FVD score increases significantly.

2. In Line 142-144, the authors stated current methods tend to require a large computation cost for training, while in the experiment section, they did not provide a quantitative comparison of their training and inference efficiency. Considering that they introduce additional trainable modules, the increase in training burden should be well discussed and reported.

3. Lacking quantitative and qualitative results on open-domain sounding video generation, such as AudioSet datasets.

**Questions:**

1. What are the textual inputs of Landscape? MM-Diffusion used non-textual inputs, does the textual inputs of the proposed method incur unfair comparison? Moreover, MM-Diffusion also reported results on A2V and V2A tasks.

2. Why does the FVD score reported in Table 2 on the Landscape dataset diverge so much from that reported in the original paper?

3.  Audio latents are 2D features and video latents are 3D features. Thus audio embeddings can be broadcast to add to video features. However, how can the video embeddings be fed to audio features? If both audio and video embeddings are 1D (temporal) features?

---

### Official Review · Reviewer_NBr8 · 2024-11-03

**Soundness:** 2
**Presentation:** 2
**Contribution:** 2
**Rating:** 3
**Confidence:** 4

**Summary:**

The authors combine base diffusion models for audio and video into a unified framework, training it to generate both modalities simultaneously. They introduce two mechanisms to enhance the alignment of audio-video pairs: 1) Timestep Adjustment, which supplies distinct timestep information to each base model; and 2) Cross-Modal Conditioning as Positional Encoding (CMC-PE),.a design for additional modules that integrates cross-modal information as temporal position data, akin to positional encoding. This approach offers an inductive bias for achieving temporal alignment in the generated data compared to the traditional cross-attention mechanism.

**Strengths:**

1. The paper proposes a unified model for joint audio and video generation, integrating base diffusion models for both modalities.It introduces two designed mechanisms, Timestep Adjustment and CMC-PE, to enhance the alignment between audio and video pairs.
2. The proposed CMC-PE method can explicitly align audio and video signal by incorporating cross-modal information like temporal position embedding. The AV-align socres in Table.1 demonstate the effectiveness of CMC-PE in improving audio-video alignment.

**Weaknesses:**

1. Lacking comparison with other sounding video generative models based on off-the-shelf audio or video generation models like [1]. The authors only reported quantitative comparison with models trained from scratch, which is unfair since the proposed method utilized pretrained models.

2. Lacking qualitative comparison between the proposed method and baseline methods. The authors only present audio-video samples generated by the proposed method. However, qualitative comparison with other methods is necessary to better explain how the proposed method can improve temporal alignment of audio and video.


[1] Xing Y, He Y, Tian Z, et al. Seeing and hearing: Open-domain visual-audio generation with diffusion latent aligners[C]//Proceedings of the IEEE/CVF Conference on Computer Vision and Pattern Recognition. 2024: 7151-7161.

**Questions:**

1. The FVD score on the Landscape dataset is much worse than that on the VGGSound dataset. However, the Landscape dataset is a quite small and domain dataset. Intuitively, Landscape should be a more simple dataset than VGGSound, thus is expected to be easier for modeling.
2. In Line 404-406, the authors claimed that the CMC-PE improves FVD than cross-attention. However, the Table 1 demonstrates poor FVD when using CMC-PE, which seems to be conflict.

---

### Official Review · Reviewer_Hy4P · 2024-11-03

**Soundness:** 3
**Presentation:** 3
**Contribution:** 2
**Rating:** 5
**Confidence:** 3

**Summary:**

The paper proposes two techniques to improve the alignment between audio-video pairs for joint generation. The timestep adjustment adjusts the difference in the noise schedules between audio and video.
Another new module, cross-modal conditioning as a positional encoding (CMC-PE), uses encoded audio as temporal position information. The experiments show improvements over some existing methods.

**Strengths:**

- The paper is clearly written.
- The idea is simple and reasonable.

**Weaknesses:**

1. My main concern is the video generation result with CMC-PE compared to cross-attention. The main advantage that the authors stated is "CMC-PE can utilize temporally local information that is suitable to temporally align the generated video with the conditional audio." However, based on Table 1, it doesn't seem to help compared to cross-attention --- The FVD score increased dramatically without the timestep adjustment. This means audio does not represent temporal position information, and adding audio information to the video feature is a distraction for video generation. It improves the alignment score, but this does not help video generation.

2. Using the timestep adjustment, the loss distribution between audio and video got closer, however, there's still a gap with $\gamma$=1.5. Does the gap get smaller with higher $\gamma$? Could you provide the Figure 2 plot for all $\gamma$ used in Table 1? I'd like to see how aligning the loss distribution reflects the generation results.

3. The comparisons with SOTA models are limited, especially more joint generation model comparisons would be better. I'm not familiar with the audio-video joint generation model literature but at least the ones mentioned in the related works could be compared like VG-VQGAN (Liu et al., 2023b), TAVDiffusion (Mao et al., 2024), and Xing et al. (2024).

**Questions:**

See the Weaknesses.

---

### Official Review · Reviewer_t6TK · 2024-11-04

**Soundness:** 3
**Presentation:** 3
**Contribution:** 3
**Rating:** 6
**Confidence:** 3

**Summary:**

This paper proposes a baseline method for sounding video generation by integrating pre-existing audio and video diffusion models, primarily training only new modules added for cross-modal alignment. The model incorporates two mechanisms to enhance the alignment of audio-video pairs: (1) timestep adjustment, which provides modality-specific timestep information to align sample generation across audio and video, and (2) Cross-Modal Conditioning as Positional Encoding (CMC-PE), where audio information is embedded akin to temporal position data and fed into the model as positional encoding.

**Strengths:**

1. The paper presents a straightforward yet effective baseline for sounding video generation.
2. Efficient use of existing video and audio models, training only additional parameters, reducing the computational cost significantly.
3. Introduces two novel and effective mechanisms, timestep adjustment and CMC-PE, which enhance the cross-modal alignment in generated data.
4. The experimental results demonstrate strong performance and validate the effectiveness of the proposed mechanisms.

**Weaknesses:**

1. Although the paper claims effective adaptation, there is a lack of experimental details regarding training efficiency, including time and hardware requirements, to substantiate this claim.
2. The paper focuses on fine-tuning new parameters only, which improves efficiency. However, it would be beneficial to see a comparison with full finetuning or LoRA finetuning to assess if performance might improve with different finetuning strategies.

**Questions:**

1. Given the critical role of high cross-modal alignment in audio-video generation, is there a quantitative measure beyond visualization to more rigorously assess alignment?

---

### Note · Authors · 2024-11-20

I have read and agree with the venue's withdrawal policy on behalf of myself and my co-authors.